# Cash transfers for HIV prevention: A systematic review

Marie C. D. Stoner [1,2]*, Kelly Kilburn [3], Peter Godfrey-Faussett [4], Peter Ghys [4], Audrey E. Pettifor [1,5,6]

1 Carolina Population Center, University of North Carolina, Chapel Hill, North Carolina, United States of America, 2 Women's Global Health Imperative, RTI International, Berkeley, California, United States of America, 3 Duke Global Health Institute, Duke University, Durham, North Carolina, United States of America, 4 UNAIDS, Geneva, Switzerland, 5 MRC/Wits Rural Public Health and Health Transitions Research Unit (Agincourt), School of Public Health, Faculty of Health Sciences, University of the Witwatersrand, Johannesburg, South Africa, 6 Department of Epidemiology, University of North Carolina, Chapel Hill, North Carolina, United States of America

* mcstoner@rti.org

**Data Availability Statement:** All relevant data are within the manuscript and its Supporting Information files.

**Funding:** This work was funded by a UNAIDS consulting award (AP). The funders had a role in

## Abstract

### Background

Given the success of cash programs in improving health outcomes and addressing upstream drivers of HIV risk such as poverty and education, there has been an increasing interest in their potential to improve HIV prevention and care outcomes. Recent reviews have documented the impacts of structural interventions on HIV prevention, but evidence about the effects of cash transfer programs on HIV prevention has not been systematically reviewed for several years.

### Methods and findings

We did a systematic review of published and unpublished literature to update and summarize the evidence around cash programs for HIV prevention from January 2000 to December 17, 2020. We included studies with either a cash transfer intervention, savings program, or program to reduce school costs. Included studies measured the program's impact on HIV infection, other sexually transmitted infections (STIs), or sexual behaviors. We screened 1,565 studies and examined 78 in full-text review to identify a total of 45 peer-reviewed publications and reports from 27 different interventions or populations. We did not do a meta-analysis given the range of outcomes and types of cash transfer interventions assessed. Most studies were conducted in sub-Saharan Africa ($N$ = 23; South Africa, Tanzania, Malawi, Lesotho, Kenya, Uganda, Zimbabwe, Zambia, and eSwatini) followed by Mexico ($N$ = 2), the United States ($N$ = 1), and Mongolia ($N$ = 1)). Of the 27 studies, 20 (72%) were randomized trials, 5 (20%) were observational studies, 1 (4%) was a case–control study, and 1 (4%) was quasi-experimental. Most studies did not identify a strong association between the program and sexual behaviors, except sexual debut (10/18 finding an association; 56%). Eight of the 27 studies included HIV biomarkers, but only 3 found a large reduction in HIV incidence or prevalence, and the rest found no statistically significant association. Of the studies that identified a statistically significant association with other STIs ($N$ = 4/8), 2

study design, decision to publish, and preparation of the manuscript.

**Competing interests:** The authors have declared that no competing interests exist.

**Abbreviations:** AOR, adjusted odds ratio; CCT, conditional cash transfer; CI, confidence interval; HSV-2, herpes simplex virus type 2; IRR, incidence rate ratio; MSM, men who have sex with men; OR, odds ratio; OVC, orphans and vulnerable children; PRISMA, Preferred Reporting Items for Systematic Reviews and Meta-Analysis; RR, risk ratio; SE, standard error; STD, sexually transmitted disease; STI, sexually transmitted infection; UCT, unconditional cash transfer.

involved incentives for staying free of the STI, and the other 2 were cash transfer programs for adolescent girls that had conditionalities related to secondary schooling. Study limitations include the small number of studies in key populations and examining interventions to reduce school costs and matched saving programs.

## Conclusions

The evidence base for large-scale impacts of cash transfers reducing HIV risk is limited; however, government social protection cash transfer programs and programs that incentivize school attendance among adolescent girls and young women show the greatest promise for HIV prevention.

## Author summary

### Why was this study done?

- Cash transfers have become a widely used policy strategy to achieve social protection and development goals in a number of different domains.

- Recent reviews have documented the impacts of cash transfer interventions on HIV prevention outcomes, but many studies have been done recently and have not yet been captured in these reviews.

### What did the researchers do and find?

- To update the current evidence related to cash transfers for HIV prevention, we did a systematic review of quantitative studies of cash transfer interventions, interventions to reduce school costs, and matched savings programs extending from January 2000 to December 2020.

- Impacts on HIV infection were mixed. Only 3 of the 8 studies that included HIV biomarkers found a reduction in HIV incidence or prevalence.

- Four of 8 studies that included other sexually transmitted infections (STIs) found a statistically significant association, and all 4 of these studies included conditionalities based on testing STI negative or secondary schooling.

- A total of 10/18 (56%) interventions identified a statistically significant reduction on delaying sexual debut, in most cases only for girls and not for boys.

### What do these findings mean?

- Overall, we find that most evidence to date is limited in demonstrating that cash transfers can reduce HIV infection or have broad reaching impacts on risky sexual behaviors.

- Social protection cash transfer programs provided to poor or vulnerable households and cash transfers conditional on school attendance were more likely to lead to delays

in sexual activity among adolescents generally and reductions in risky sex among adolescent girls, at least while the programs were ongoing.

- Further research is needed to understand the impact of cash transfer among key populations and when combined with other HIV prevention interventions.

## Introduction

Globally, cash transfers have become one of the most popular policy strategies to achieve social protection and development goals in a number of different domains. Programs that provide noncontributory cash payments now reach over 1 billion people across more than 130 countries [1]. Evidence from these programs consistently points to their positive impacts on monetary poverty, education, health and nutrition, productivity and employment, and empowerment [1]. Alongside large, national programs, there are a range of other initiatives and interventions that have utilized cash payments to achieve specific outcomes, many of them either education or health based. Given the success of cash programs in improving health outcomes and addressing upstream drivers of HIV risk such as poverty and education, there has been an increasing interest in their potential to improve HIV prevention and care outcomes.

### Background on cash transfer interventions

In this review, we focus on several models of cash transfers that have been used to prevent HIV including cash transfer programs (government programs and stand-alone/research studies), interventions to reduce school costs (scholarship, school uniform, or school fee), and matched savings programs. Cash transfer programs mainly fall into 2 categories: (1) cash payments to poor families with the aim of poverty alleviation and social protection; and (2) cash payments as incentives for behavior change [2]. The first group of programs is based on the theory that cash payments can be used to improve underlying structural factors related to HIV risk. National government run cash transfer programs fall into the first category because they are designed to transfer cash to poor households with the goal of helping families meet their basic needs such as food consumption, housing, and healthcare. The second type of cash transfer scheme is based on the theory that cash can be used as an incentive to promote behavior change. These programs use cash transfers as incentives for individuals to engage in protective behaviors or remaining HIV or STI negative [2,3].

Cash transfer programs can have unconditional or conditional designs. Unconditional cash transfers (UCTs) provide cash assistance to individuals or households without any obligations and therefore seek to encourage behaviors through a change in income resulting in changes in the demand for services. Conditional cash transfers (CCTs), on the other hand, explicitly condition the receipt of cash payments on certain behaviors that are deemed beneficial such as school attendance or healthcare utilization and track compliance. Incentive-based interventions are designed to reduce risk by providing immediate benefits for avoiding high-risk behaviors [2,3]. For example, individuals at risk of HIV must balance immediate benefits of risky sex with the long-term costs of possible HIV infection. Cash incentives are designed to provide a "nudge" to avoid immediate gratification of certain behaviors by providing an incentive to not engage in those high-risk behaviors [3,4].

## Current evidence on cash transfers for HIV prevention

Recent reviews have documented the impacts of cash transfer interventions on HIV prevention for young women [5] and household economic strengthening for HIV outcomes [6]. Evidence from previous reviews suggest that some of the strongest impacts of cash transfers are on health and schooling outcomes for the poor. The largest systematic review of cash transfer programs to date finds that across countries, cash transfers increase the consumption of diverse foods, improve the use of health services, and increase school attendance [1]. Additionally, evidence indicates that cash transfers can reduce anxiety and stress, improve self-esteem and hope for the future, and reduce early marriage and pregnancy in adolescents [7]. A recent review of conditional incentive interventions found that in the short term, these interventions can increase HIV testing rates, increase voluntary male circumcision, and improve other HIV prevention and treatment outcomes in certain settings, but results are not maintained after the study ends [8].

The aim of this review was to update the current evidence related to cash transfers for HIV prevention. Here, we use reviews, published, and unpublished literature to summarize the evidence around cash transfers and payments for HIV prevention behaviors. We reviewed quantitative studies of cash transfer interventions, interventions to reduce school costs, and matched savings programs examining HIV infection and sexual behavior outcomes in any population.

## Methods

This systematic review was conducted utilizing the standard protocol for Preferred Reporting Items for Systematic Reviews and Meta-Analysis (PRISMA; PRISMA checklist is available in S1 PRISMA Checklist) [9]. We included quantitative studies of cash transfer interventions, interventions to reduce school costs, and matched savings programs extending from January 2000 to December 2020. Searches were done on July 25, 2019, and December 17, 2020. We categorized cash transfer interventions as either government social protection programs, individual incentive-based cash transfer programs, or individual structural cash transfer programs to alleviate poverty. We reviewed peer-reviewed literature, gray literature (e.g., reports, working paper, etc.), and ongoing studies where data were available.

We first employed a systematic literature review to find all published studies that met our review criteria and were in English. We did another search on October 26, 2021 to update our systematic review to also include articles not in English from the same period of January 2000 to December 2020. Studies that met inclusion criteria were those that (1) analyzed either cash transfer programs, savings program, or programs to reduce school costs; and (2) reported impacts on HIV and HIV prevention–related sexual behavior outcomes. The outcomes of interest were HIV infection (incidence and prevalence), other sexually transmitted infections (STIs) (all infections; incident or prevalent), condom use, sexual debut, number of partners, transactional sex, older partners, and other behavioral outcomes related to these sexual behaviors such as combined behavioral risk scores. Our search criteria included cash transfer, cash incentive, financial incentive, cash reward, monetary reward, contingency management, savings, scholarship, school uniform, school fee, or uniform costs and HIV, sexually transmitted disease (STD), STI, condom use, sexual debut, number of partners, sexual partners, transactional sex, older partner, or sexual behavior (see S1 Text for full search terms for each database). We searched the databases PsycINFO, EconLit, PubMed, and Web of Science. We did not do a meta-analysis because the group of studies assessed was not sufficiently homogenous; the studies have both a large variety of outcomes and various types of cash transfer

interventions assessed. The study protocol was prepared on February 4, 2019 for UNAIDS but was not registered in any publicly accessible database (S2 Text).

Abstracts were imported from each database and combined into Covidence online software. Two reviewers (MCDS and KK) screened the abstracts independently and then examined the full text of articles that met the criteria or were flagged by the software as discrepancies between reviewers. Discrepancies were reviewed again by the 2 reviewers to come to a joint decision. Inclusion was based on agreement between the 2 reviewers. We identified additional studies in the gray literature by contacting experts in the field and from other systematic reviews or commentaries on the topic. One of the reviewers (MCDS) extracted information from all included studies including the study population, timeline, location, type of intervention, research design, conditionalities, and effect sizes with confidence intervals (CIs) or standard errors (SEs) and significance for each of the relevant outcomes reported (S1 Table). We did not restrict studies by type of estimate that was reported (e.g., odds ratio (OR)). We assessed study quality by examining the strength of the research design and sample size.

## Results

A total of 1,642 records were imported through the database search, and 8 additional studies were added from the gray literature (Fig 1). Of these studies, 1,607 were screened and 1,484 records were excluded. In the full-text review, 78 studies were included, and 33 were excluded that did not meet inclusion criteria. We identified a total of 45 peer-reviewed publications and reports from 27 different interventions or populations (Table 1) [10–55]. Most studies were conducted in sub-Saharan Africa ($N$ = 23; South Africa, Tanzania, Malawi, Lesotho, Kenya, Uganda, Zimbabwe, Zambia, and eSwatini) followed by Mexico ($N$ = 2), the US ($N$ = 1), and Mongolia ($N$ = 1)). Of the 27 studies, 20 (74%) were randomized trials, $N$ (18%) were observational studies, 1 (4%) was a case–control study, and 1 (4%) was quasi-experimental. Most interventions assessed sexual behavior outcomes in adolescent populations ($N$ = 15; 3 among girls only), followed by orphans and vulnerable children (OVC) ($N$ = 5), both adolescents and adults ($N$ = 2), adults only ($N$ = 1), men who sell sex ($N$ = 1), and women who sell sex ($N$ = 1). Of the studies among adolescents, most studies (8 of 14) involved a government cash transfer to the household. We identified 12 ongoing studies (S2 Table).

A total of 21 studies included a cash transfer intervention (33 publications or reports) (Table 1). Three studies (4 publications or reports) assessed a matched savings program. Three interventions (8 publications or reports) assessed interventions to reduce school costs. Of these 21 cash transfer interventions, 9 studies assessed the impact of household grants from the government, 4 were incentive-based programs, and 8 were individual cash transfers for poverty alleviation. Government cash transfer programs/grants were assessed in Kenya, Zambia, Malawi, Tanzania, Zimbabwe, and in South Africa where 4 studies assessed different programs (child support grant or foster grant ($N$ = 1); child support grant alone ($N$ = 2); and old-age pension ($N$ = 1)).

### HIV outcomes

Eight studies examined HIV incidence of prevalence as an outcome, and results were mixed (Table 2; S1 Table). Three of the 5 cash transfer studies that assessed HIV infection found a significant reduction; a cash transfer intervention provided to young women aged 13 to 22 years, and their families, to stay in school in Malawi found a reduction in HIV prevalence (adjusted odds ratio [AOR] 0.36; 95% CI: 0.14, 0.91 [10]), and a lottery intervention for adults who tested STI negative in Lesotho found a reduction in HIV incidence by 2.5 percentage points (95% CI: 0.0%, 5.0%, $p$ = 0.046) and HIV prevalence by 3.4 percentage points (95% CI: 0.0%, 5.9%;

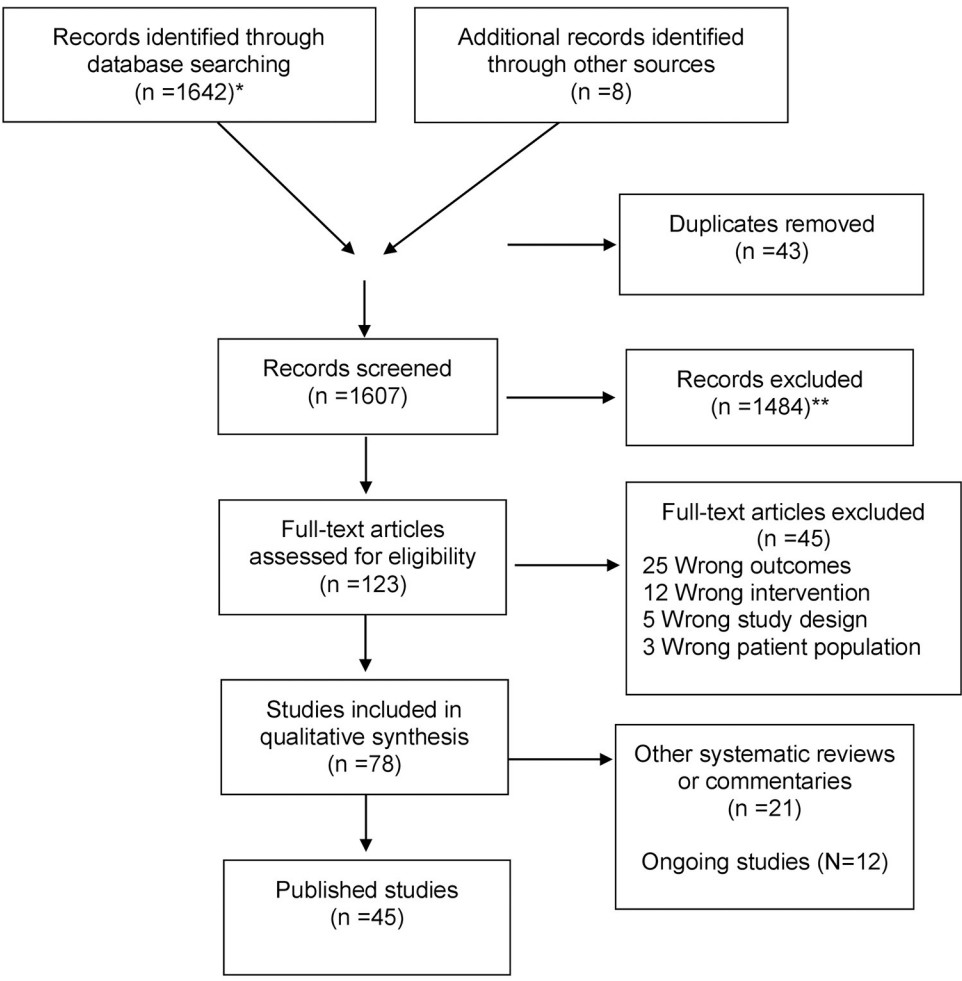

*Includes English and non-English articles
**Studies were excluded that 1) did not include either cash transfer programs, savings program, or programs to reduce school costs, 2) did not include measures of HIV, other STIs, or sexual behaviors, 3) were qualitative studies, or, 4) were reviews or conference abstracts.

**Fig 1. PRISMA flowchart for inclusion and exclusion criteria.** PRISMA, Preferred Reporting Items for Systematic Reviews and Meta-Analysis; STI, sexually transmitted infection.

$p = 0.044$) [33]. An intervention for adolescent girls combining incentives to enroll and attend school with a lottery conditional on being STI negative in eSwatini reduced HIV incidence among girls getting educational incentives (OR 0.77; 95% CI 0.60 to 0.98) and with the STI lottery alone (OR 0.83 95% CI 0.65 to 1.07,[19,56]), although the lottery alone was not significant. Conversely, 2 other cash transfer studies did not reduce HIV incidence: one study examining the effect of a cash transfer conditional on school attendance among adolescent girls in South Africa (AOR 1·17; 95% CI: 0·80 to 1·72; $p = 0.42$) [37] and another examining an incentive conditional on testing HIV negative ($\beta = 0.001$, robust SE, 0.005) [29]. One study that examined a cash transfer that had various school-based conditionalities did not report the impact on HIV incidence in boys and girls in South Africa due to low HIV incidence ($N = 75$) [28]. Two studies that evaluated programs to reduce school costs for adolescent orphans in Kenya

**Table 1. List of included studies by population, location, age and sex, category, sample size, and study design.**

| # | Citation | Target population | Location | Population: Age and sex | Category | Sample size | Study design | Timeline |
|---|---|---|---|---|---|---|---|---|
| 1 | Handa 2014; Rosenberg 2014; Handa 2017 | OVC: HH | Kenya | Females and males; age 15 to 25 | Government cash transfer (UCT) | 2,210 HH; 1,429 youth | Cluster RCT | 2007 to 2011 |
| 2 | Cluver 2013; Cluver 2014; Cluver 2017 | Adolescents: HH | South Africa | Females and males; age 12 to 18 | Government cash transfer (UCT) | 2,688 | Case control | 2009 to 2012 |
| 3 | Heinrich 2017 | Adolescents: HH | South Africa | Females and males; age 15 to 17 | Government cash transfer (UCT) | 1,531 | Observational | 2010 to 2011 |
| 4 | Siaplay 2012 | Adolescents: HH | South Africa | Females and males; ages 14 to 26 | Government cash transfer (UCT) | 3,307 | Observational | 2002 to 2006 |
| 5 | AIR 2015 | Adolescents: HH | Zambia | Households with OVC; Females and males; 13 to 17 at baseline | Government cash transfer (UCT) | 3,077 HH | RCT | 2011 to 2014 |
| 6 | Malawi Social Cash Transfer Program Evaluation Team 2016 | Adolescents: HH | Malawi | Females and males; 13 to 19 | Government cash transfer (UCT) | 3,521 HH; 2,325 youth at endline | RCT | 2013 to 2015 |
| 7 | UNICEF 2018 | Adolescents: HH | Tanzania | Females and males; 14 and 28 years at baseline | Government cash transfer (UCT) | 1,397 youth | Cluster RCT | 2015 to 2017 |
| 8 | DSD, SASSA, and UNICEF 2012 | Adolescents: HH | South Africa | Females and males; 15 to 17 year olds | Government cash transfer (UCT) | 1,231 adolescents | Observational | 2010 |
| 9 | Zimbabwe Harmonised Social Cash Transfer Evaluation Team 2018; AIR 2014 | Adolescents: HH | Zimbabwe | Females and males; 13 to 24 years | Government cash transfer (UCT) | 2,567 households; 2,310 youth | Quasi-experimental | 2013 to 2017 |
| 10 | Galárraga 2017; Galárraga 2014 | Male sex workers | Mexico | Male sex workers age 18 to 40; MSM | Incentive-based individual cash transfer (both UCT and CCT) | 227 | RCT | 2012 to 2014 |
| 11 | De Walque 2012 (PR); Cooper 2018; Packel 2010 | Adults | Tanzania | Females and males; 18 to 30 years | Incentive-based individual cash transfer (CCT) | 2,399 | RCT | 2009 to 2011 |
| 12 | Nyqvist 2018 | Adults | Lesotho | Females and males; age 18 to 32 | Incentive-based individual cash transfer (CCT) | 3,029 | RCT | 2010 to 2013 |
| 13 | Kohler 2011 | Adolescents and adults | Malawi | Females and males; 16 to 75 years | Incentive-based individual cash transfer (CCT) | 1,307 individuals | RCT | 2006 |
| 14 | Ssewamala 2010; Jennings 2015 | OVC | Uganda | Female and males; 10 to 17 years | Savings | 346 | Cluster RCT | 2005 to 2008 |
| 15 | Witte 2015 | Female sex workers | Mongolia | Females age 18 + involved in street-based sex work | Savings | 107 | Group RCT | 2011 to 2013 |
| 16 | Cho 2011; Cho 2017; Cho 2018 | OVC | Kenya | Females and males; 12 to 14 years | School costs | 835 | Cluster RCT | 2008 to 2009 |
| 17 | Hallfors 2015; Halifors 2011; Luseno 2015 | OVC | Zimbabwe | Orphan girls in grade 6 at baseline | School costs | 328 | Cluster RCT | 2007 to 2010, 2011 to 2012 |
| 18 | Duflo 2006; Duflo 2015 | Adolescents | Kenya | Grade 6 at enrollment (both boys and girls) | School costs | 10,651 youth, 328 schools | Cluster RCT | |
| 19 | Baird 2012; Baird 2010; Baird 2017; Beauclair 2018 | Adolescents | Malawi | Females 13 to 22 years | Structural individual cash transfer (both UCT and CCT) | 1,289 | Cluster RCT | 2008 to 2009 |
| 20 | Minnis 2014 | Adolescents | US | Females and males 16 to 21, self-identify as Latino | Structural individual cash transfer (CCT) | 162 | Social network randomized RCT | 2010 to 2012 |

(*Continued*)

**Table 1.** (Continued)

| # | Citation | Target population | Location | Population: Age and sex | Category | Sample size | Study design | Timeline |
|---|----------|-------------------|----------|------------------------|----------|-------------|--------------|----------|
| 21 | Pettifor 2016 | Adolescents | South Africa | Females 13 to 20 | Structural individual cash transfer (CCT) | 2,533 | RCT | 2011 to 2013 |
| 22 | Goodman 2014; Goodman 2016 | OVC: headed households | Kenya | OVC-headed households (aged 13 to 25); Females and males | Structural individual cash transfer (CCT) | 1,060 | Observational | 2012 to 2014 |
| 23 | Galárraga 2009 | Adolescents | Mexico | Females and males; 12 to 24 years of age | Structural individual cash transfer (CCT) | 3,743 individuals | Observational | 2004 |
| 24 | Karim 2015 | Adolescents | South Africa | Females and males; 13 years of age and older Grade 9 to 10 | Structural individual cash transfer (CCT) | 2,949 youth | School matched pair cluster randomized control trial | 2010 to 2012 |
| 25 | Görgens 2019 | Adolescents | eSwatini | Females age 15 to 22 at baseline | Structural individual cash transfer (CCT) and incentive-based individual cash transfer (CCT) | 4,329 | 2 × 2 factorial randomized trial with 4 intervention groups | 2016 to 2019 |
| 26. | Schaefer 2020 | Adolescents and adults | Zimbabwe | Females and males age group 15 to 29 years and aged 30 to 54 years | Structural household cash transfer (CCT and UCT) | 3,516 | Linked cohort data to data form 3-arm cluster RCT | 2010 to 2011 |
| 27 | Moscoe 2019 | Adults | Kenya | Adult men over 21 years old | Savings (prize linked) | 300 | RCT | 2018 |

Studies from the same population and intervention were grouped together as one intervention.

CCT, conditional cash transfer; HH, grant provided to the household; MSM, men who have sex with men; OVC, orphans and vulnerable children; RCT, randomized controlled trial; UCT, unconditional cash transfer.

and Zimbabwe also found no significant effect on HIV prevalence, although there was a non-significant reduction in one study (AOR 0.72; 95% CI: 0.15 to 3.42; $p = 0.68$ and AOR 1.15; 95% CI: 0.47 to 2.79) [21,52]. None of the governmental cash transfer intervention studies or savings program studies evaluated the impacts of the programs on HIV or other STIs.

## Sexually transmitted infections outcomes

Eight studies examined other STIs. Five studies examined herpes simplex virus type 2 (HSV-2) infection, but only 2 of these 5 studies found a statistically significant reduction. One study found a reduction in HSV-2 incidence with a cash transfer intervention conditional on various schooling-related outcomes in both adolescent girls and boys combined (incidence rate ratio (IRR): 0.70, 95% CI: 0.57 to 0.86, $p = 0.007$; [28]), while the other cash transfer trial conditional on school attendance among girls in Malawi reduced prevalence of HSV-2 but not of syphilis (HSV-2 OR 0.24, 95% CI: 0.09 to 0.65; syphilis AOR 0.91; 95% CI: 0·12 to 6·8 [10]). In addition, 3 incentive-based studies examined composite measures of STIs. All 3 studies were cash transfer programs conditional on negative STI status. One of these, an incentive-based intervention conditional on testing negative for 4 curable STIs in adults in Tanzania, identified a reduction in incidence of a composite measure that included *Chlamydia trachomatis*, *Neisseria gonorrhoeae*, *Trichomonas vaginalis*, *and Mycoplasma genitalium* (risk ratio (RR): 0.73, 95% CI: 0.47 to 0.99; [57]). However, the same study did not find a reduction in a different composite measure that included prevalence of HIV, HSV-2, or syphilis (RR 1.03, 95% CI: 0.74 to 1.32; [57]). Nyquist and coauthors also found that a lottery incentive for adults in Lesotho that was conditional on testing negative for syphilis and trichomoniasis led to a reduction in

**Table 2. Breakdown of the effects of cash transfer, saving and school costs intervention studies on HIV infection and related sexual behaviors.**

| Intervention | Study population | Location | Associated studies | HIV infection | Unprotected sex | Sexual debut | Number of partners | Transactional sex | Older partners | Other STIs | Other |
|---|---|---|---|---|---|---|---|---|---|---|---|
| Incentive-based individual cash transfer | Females and males, ages 18 to 32 | Lesotho | Nyqvist, 2018 | Neg | Neg | - | - | - | - | Neg | - |
| | Females and males, ages 16 to 75 | Malawi | Kohler, 2011 | Null | Null ♀ Pos ♂ | Neg ♀ Pos ♂ | - | - | - | Neg (STI composite) Null (STI composite with HIV) | Neg ♀ Pos ♂ Measure: combined sex and condom use |
| | Male sex workers ages 18 to 40 | Mexico | Galárraga 2017; Galárraga 2014 | - | Neg | - | Null | - | - | Null | - |
| | Females and males, ages 18 to 30 | Tanzania | De Walque 2012; Cooper 2018; Packel 2010 | - | - | - | - | - | - | - | Null (overall) Neg ♀ Measure: behavior change |
| | Females, ages 13 to 22 | Malawi | Baird 2012; Baird 2010; Baird 2017; Beauclair 2018 | Neg | Null | Null | - | - | Neg | Neg (HSV-2) Null (Syphilis) | Neg Measure: sexual intercourse once per week |
| | Females, ages 15 to 22 | eSwatini | Görgens 2019 | Neg | - | - | - | - | - | - | - |
| | Females and males ages 16 to 21, self-identify as Latino | US | Minnis 2014 | Null | Null | - | - | - | - | - | - |
| | Females, ages 13 to 20 | South Africa | Pettifor 2016 | Null | Neg | Null | Neg (any sex partner) Null (>1 sex partner) | Null | Null | Null (HSV-2) | - |
| | Females and males; 12 to 24 years of age | Mexico | Galárraga 2009 | - | Null | Null | - | - | - | - | - |
| | OVC-headed households (aged 13 to 25); Females and males | Kenya | Goodman 2014; Goodman 2016 | - | Neg ♀ Pos ♂ | Neg ♀ Pos ♂ | Null | - | - | - | - |
| | Females and males, 13 years of age and older, Grade 9 to 10 | South Africa | Karim 2015 | - | - | - | - | - | - | Neg (HSVNeg2) | - |
| | Females and males aged 15 to 29 and 29 to 54 | Tanzania | Schaefer 2020 | - | Null | Null | Null | - | - | - | - |

*(Continued)*

**Table 2.** (Continued)

| Intervention | Study population | Location | Associated studies | HIV infection | Unprotected sex | Sexual debut | Number of partners | Transactional sex | Older partners | Other STIs | Other |
|---|---|---|---|---|---|---|---|---|---|---|---|
| National social protection programs | Females and males, ages 15 to 25 | Kenya | Handa 2014; Rosenberg 2014; Handa 2017 | - | Null | Neg | Null | Null | Null | - | - |
| | Females and males, ages 12 to 18 | South Africa | Cluver 2013; Cluver 2014; Cluver 2017 | - | Null | - | Null ♀ Neg ♂ | Neg ♀ Null ♂ | Neg ♀ Null ♂ | - | Neg Measure: composite risk score |
| | Females and males, ages 14 to 26 | South Africa | Siaplay 2012 | - | Null | Neg ♀ Null ♂ | Null | - | - | - | - |
| | Females and males, ages 13 to 17 | Zambia | AIR 2015 | - | Null | Null | Null | Null | Neg | - | - |
| | Females and males, ages 13 to 19 | Malawi | Malawi Social Cash Transfer Program Evaluation Team 2016 | - | Null | Neg (midline) Null (endline) | Null | Null | Neg | - | - |
| | Females and males, 14 and 28 years | Tanzania | UNICEF 2018 | - | Null | Null | Null | Null | Null | - | - |
| | Females and males, ages 15 to 17 | South Africa | DSD, SASSA, and UNICEF 2012 | - | - | Neg | - | - | - | - | - |
| | Females and males, ages 13 to 24 | Zimbabwe | Zimbabwe Harmonised Social Cash Transfer Evaluation Team 2018; AIR 2014 | - | Null | Neg | Null | - | Null | - | - |
| | Females and males, ages 15 to 17 | South Africa | Heinrich 2017 | - | - | Neg ♀ Null ♂ | Neg ♀ Null ♂ | - | - | - | - |
| Savings | Females and males, ages 18 and older, involved in street-based sex work | Mongolia | Witte 2015 | - | Null | - | - | - | - | - | Neg (sex acts for pay) |
| | Females and males, ages 10 to 17 | Uganda | Ssewamala 2010; Jennings 2015 | - | - | - | - | - | - | - | Neg (HIV-preventive attitudinal scores) |
| | Adult men over 21 | Kenya | Moscoe 2019 | - | - | - | - | Null | - | - | - |
| School costs | Females and males, ages 12 to 14 | Kenya | Cho 2011; Cho 2017; Cho 2018 | Null | Null | Null | Null | Neg | - | Null (HSV-2) | - |
| | Females, grade 6 (ages 10 to 16) | Zimbabwe | Hallfors 2015; Hallfors 2011; Luseno 2015 | Null | Null | Neg | - | - | - | Null (HSV-2) | - |

(*Continued*)

**Table 2.** (Continued)

| Intervention | Study population | Location | Associated studies | HIV infection | Unprotected sex | Sexual debut | Number of partners | Transactional sex | Older partners | Other STIs | Other |
|---|---|---|---|---|---|---|---|---|---|---|---|
| | Females and males, grade 6 (average age of 14) | Kenya | Duflo 2006; Duflo 2015 | - | - | Neg ♀ Null ♂ | Null | - | - | - | - |

"Null" indicates no significant effect, Neg indicates significant decrease, and Pos indicates significant increase (♀ separate effect on women ♂ separate effect on men);— is not assessed.

HSV-2, herpes simplex virus type 2; OVC, orphans and vulnerable children; STI, sexually transmitted infection.

prevalence of both syphilis and trichomoniasis by 3.2 percentage points (95% CI: 1.4%, 5.0%; $p < 0.001$) [33].

## Sexual behavior outcomes

A total of 19 studies examined unprotected sex as an outcome, 15 examined partner number, 8 examined transactional sex, and 8 examined partner age. Among the studies that examined the impact of cash transfers on sexual behavior outcomes, most found a statistically insignificant association including on unprotected sex ($N = 14/19$), partner number ($N = 11/15$), transactional sex ($N = 6/8$), and having an older partner ($N = 4/8$). Among studies that reported differences disaggregated by biological sex, several studies found larger reductions among women or girls compared to men or boys.

Interventions that did have an overall statistically significant reduction in unprotected sex, partner number, transactional sex, and having an older partner varied widely in the type of intervention and population studied (S1 Table). The 3 interventions that found statistically significant reductions in unprotected sex were all cash transfer programs but among diverse populations including a cash transfer conditional on school attendance in adolescent girls (RR 0.81, 95% CI: 0.67 to 1.0; [37]); an incentive-based intervention conditional on testing STI negative among men who sell sex (high incentive $\beta = 0.113$, SE = 0.060, $p = <0.10$; [16]) and an incentive-based lottery for testing STI negative among adults in Lesotho ($\beta = 10.85$; SE = 5.89, $p < 0.10$) [33]. The South African cash transfer intervention conditional on schooling reduced the risk of having any sex partner in the last 12 months (RR 0.90, 95% CI: 0.83 to 0.99), but did not significantly reduce having more than 1 sex partner in the last 12 months (RR 0.86 (95% CI: 0.67 to 1.1) [37]. One study identified a statistically significant reduction in transactional sex examining the association with a household government grant program among adolescents in South Africa (AOR 0.49, 0.25 to 0.96, $p = 0.03$; [53]). Additionally, a savings program for women (aged 18 or older) involved in street-based sex work led to a statistically significant reduction in the number of unprotected vaginal sex acts for pay (IRR 0.78, 95% CI: 0.67 to 0.71); however, it is important to note that this is a different indicator than transactional sex in the other studies [40]. Of the 3 studies that found an impact on having an older partner, 2 were among adolescents in households receiving government cash transfers in Malawi (−3.9 percentage points, $p < 0.05$) and Zambia (−3.3 percentage points, $p < 0.01$) [41,46]. The third program was a cash transfer conditional on school attendance for female adolescents in Malawi (AOR 0.21, 95% CI: 0.07 to 0·60; [10]) [37].

In contrast to other sexual behaviors, most studies examining sexual debut found a statistically significant reduction in ever having sex or a delay in age of first sex ($N = 10/18$), although some of them identified a reduction for only girls or women but not boys or men ($N = 6/18$).

Of the 10 interventions that saw a statistically significant reduction even if the reduction was only in females, one was a cash transfer conditional on completion of life skills activities among both adolescent males and females in the US [30], 5 were national government cash transfer programs [23,26,45,47,49], 1 was a study CCT to OVC-headed households in Kenya [18],and 2 were programs to reduce school fees adolescents in Kenya [15] and OVC in Zimbabwe [21] (see estimates in S1 Table). The majority of national government cash transfer programs decreased the proportion with early sexual debut [23,45,49], although 2 only had a reduction in girls but not boys [26,47].

### Long-term impacts and combination impacts

Three studies examined the longer-term impact of cash interventions once the programs end, and findings suggest that impacts are not sustained once the cash stops. The interventions that were evaluated include the Malawi Zomba cash transfer for adolescent girls [43], the incentive-based intervention among men who sell sex in Mexico [16], and an intervention to reduce school costs for girls in Kenya [15]. Yet, recent evidence suggests that cash transfers may be more effective in combination with other interventions and may also lead to longer-term impacts [54,56,58]. For example, in Kenya, subsidies alone did not have a lasting impact on HSV-2 in girls or boys 7 years following the implementation of an education subsidy program, but the program did reduce HSV-2 in girls when subsidies were combined with the government's HIV curriculum [15]. Lastly, an observational study in South Africa found that receipt of a government grant in combination with teacher or parental support was associated with a larger reduction in HIV risk behavior than with receipt of cash alone [54].

### Ongoing studies

We identified 12 ongoing studies: 10 of these studies are being conducted in adolescent populations, 9 of which focus on adolescent girls and young women (S2 Table) [59–69]. All 8 of the interventions include adolescents are in sub-Saharan Africa (Ghana, South Africa ($N$ = 4), Zimbabwe, Kenya ($N$ = 2), Malawi, Uganda, and Tanzania). The 2 other studies are with women who sell sex in Kazakhstan [64] and Uganda [66]. All studies are combination prevention studies combining cash assistance with other interventions.

## Discussion

In this review of cash transfer programs for HIV prevention, we found that overall, there is limited evidence for the impact of cash transfers for reducing new HIV infections. The strongest evidence that emerged was for HIV prevention behavior change, specifically delaying sexual debut for young people; 10/18 (56%) interventions had a statistically significant reduction on delaying sexual debut, in a number of cases only for girls and not for boys. For the majority of other HIV risk behaviors examined including partner number, older partners, and transactional sex, the evidence is not as strong—about a third of the studies found a reduction in risky behaviors.

It is noteworthy that government social protection programs, which target the most poor and vulnerable households, have shown some of the strongest impacts on HIV risk reduction, particularly among adolescents. The majority of program evaluations looking at HIV prevention have been among adolescents and young people. As government social protection cash transfers intend to reduce poverty and smooth consumption, the evidence of their impact on adolescent behavior supports the hypothesis that addressing upstream drivers of HIV risk such as poverty can reduce risky behavior, particularly for the most vulnerable. In addition, there is a strong evidence base that cash transfers (both conditional and unconditional programs) can

improve school enrollment and attendance and that schooling is protective for HIV infection. Cash transfer programs, therefore, may reduce risk the most by keeping adolescents in school, particularly girls. In fact, most of the positive impacts on adolescents' risky sexual behavior are among girls; associations among boys were often null or in the opposite direction. This is in line with the literature on the impact of cash transfers to improve school enrollment and attendance where effects are stronger for girls than boys [5]. The one trial to date showing an impact of cash transfers on HIV incidence was an explicit incentive paid to adolescent girls for enrolling in and attending school in eSwatini where secondary school enrollment for girls is generally low (33%) [56].

One of the other major mechanisms through which cash transfers are thought to reduce HIV risk for girls and young women are by reducing girl's financial dependence on male partners and thus reducing the need for sexual partnerships that include transactional sex [70]. However, of the 8 studies that examined transactional sex, only 2/8 (25%) found a statistically significant reduction—one was among young women in South Africa living in homes receiving the child support grant [53] and the second was in Kenya among OVC receiving support for schooling costs [52]. Additionally, there is emerging evidence that context may be important in determining the impact of cash transfers on reducing risk behavior. In settings where transactional sex is driven primarily by basic needs (e.g., obtaining food), small cash transfers may have an impact on reducing risk for poor young women [71]. In settings where the primary motivator for engagement in transactional sex is related to obtaining material goods to increase social status and self-esteem, it is less likely that small cash payments will have much of an impact. There is evidence that engagement in transactional sex is associated with low self-esteem, and thus programs that combine cash with other program elements to increase hope for the future and self-esteem may have more promise than cash alone [72].

While all studies reported on at least 1 HIV risk behavior, not many collected HIV biomarkers, so evidence is limited to make conclusions on the direct impacts of cash transfers on HIV. Eight of the 27 studies included HIV biomarkers, but only 3 found a statistically significant reduction in HIV incidence or prevalence, and the rest found no impact. The 3 studies that showed an impact included a cash transfer conditional on schooling among adolescent girls in Malawi (reduction in HIV prevalence), an incentive-based lottery study among adults in Lesotho for testing negative for STIs (reduction in HIV incidence), and a cash transfer conditional on school enrollment and attendance in eSwatini (reduction in HIV incidence). There was slightly more evidence of the impact of cash transfers on STI outcomes. Of the studies that found an impact on STIs (N = 4/8), 2 involved incentives for staying free of the STI, and the other 2 were cash transfer programs for adolescent girls that had conditionalities related to secondary schooling.

The evidence of the impact of cash transfers among key populations is limited. Of the 27 studies included in this review, 1 was among men who sell sex and who had sex with men in Mexico and 1 study was among women who sell sex in Mongolia. Only 2 of the 12 ongoing studies is being conducted among key populations, interventions for women who sell sex in Kazakhstan and Uganda. More studies are needed to evaluate the impacts of cash transfer interventions and government grants among key populations who may experience discrimination or have limited access to services and may benefit from these programs.

There are 2 additional areas where more research is needed. First, there is very limited evidence to date that combining social protection programs with additional support (e.g., caring adult and other social support services) can have stronger impacts on reducing HIV risk behaviors. In this area, however, there are at least 3 large programs in the field (DREAMS, Global Fund, and Tanzania/UNICEF) that should produce more evidence about the impact of combination programs in reducing HIV risk. Second, there is limited evidence about the

impact of cash transfer programs for either out-of-school girls or young women aged 18 to 24 year where HIV incidence is highest. In the "Zomba trial" that included out-of-school girls, the prevalence of most risk behaviors was too low at baseline to see any significant difference at the end of the program [10]; however, some of the DREAMS programs include out-of-school girls: The Sauti program in Tanzania and the AGI-Kenya study, implemented by the Population Council, include out-of-school girls.

This review focuses on several different forms of interventions to provide cash payments, some explicitly for HIV prevention and some intended for poverty reduction. Overall, there are a large number of studies evaluating cash transfers and national government social protection cash transfer programs with large sample sizes and rigorous methods (and the majority are randomized controlled trials). However, the risk of bias in the study outcomes was difficult to evaluate across studies because of the wide range of intervention types and targeted populations, but the strong study designs of the majority of evaluations suggest that the evidence generated to date is robust. Given that we were unable to do a meta-analysis, studies are summarized by statistical significance and should also be considered with the estimates in S1 Table. The main area where evidence is still lacking is for specific populations (e.g., people who sell sex and men who have sex with men [MSM]) and for transactional sex which has not been evaluated in many studies. There is less evidence on interventions to reduce school costs ($N$ = 3) and matched saving programs ($N$ = 3) or combination interventions that add multiple elements into a single program (cash plus programs). Few interventions that have actually measured HIV incidence.

Overall, we find that most evidence to date is limited in demonstrating that cash transfers can reduce HIV infection or have broad reaching impacts on risky sexual behaviors. However, there are some populations and program designs that seem to be more promising for impacting HIV preventive behaviors. Social protection cash transfer programs provided to poor or vulnerable households and cash transfers conditional on school attendance (or related to incentivizing schooling attendance for girls) were more likely to lead to delays in sexual activity among adolescents generally and reductions in risky sex among adolescent girls, at least while the programs were ongoing. To date, the strongest evidence related to cash transfer programs for HIV prevention suggests that social protection programs for poor and vulnerable families may reduce risk behaviors of adolescents living in those homes, especially girls. Program and policymakers interested in HIV prevention for young women should consider programs that directly incentivize school enrollment and attendance or are conditional on attendance, which may have the largest impact on HIV risk for girls, especially in contexts where secondary school attendance is low.

## Supporting information

**S1 PRISMA Checklist. PRISMA Checklist.** PRISMA, Preferred Reporting Items for Systematic Reviews and Meta-Analysis.
(DOCX)

**S1 Table. List of included studies.**
(XLSX)

**S2 Table. List of ongoing studies.**
(XLSX)

**S1 Text. Search terms.**
(DOCX)

**S2 Text. Protocol.**
(PDF)

## Author Contributions

**Conceptualization:** Marie C. D. Stoner, Kelly Kilburn, Peter Godfrey-Faussett, Peter Ghys, Audrey E. Pettifor.

**Data curation:** Marie C. D. Stoner, Kelly Kilburn.

**Formal analysis:** Marie C. D. Stoner, Kelly Kilburn.

**Funding acquisition:** Peter Godfrey-Faussett, Peter Ghys, Audrey E. Pettifor.

**Investigation:** Marie C. D. Stoner, Kelly Kilburn, Audrey E. Pettifor.

**Methodology:** Marie C. D. Stoner, Kelly Kilburn, Peter Godfrey-Faussett, Audrey E. Pettifor.

**Project administration:** Audrey E. Pettifor.

**Resources:** Peter Godfrey-Faussett, Peter Ghys, Audrey E. Pettifor.

**Software:** Marie C. D. Stoner, Kelly Kilburn.

**Supervision:** Audrey E. Pettifor.

**Validation:** Marie C. D. Stoner, Kelly Kilburn.

**Visualization:** Marie C. D. Stoner.

**Writing – original draft:** Marie C. D. Stoner, Kelly Kilburn, Audrey E. Pettifor.

**Writing – review & editing:** Marie C. D. Stoner, Kelly Kilburn, Peter Godfrey-Faussett, Peter Ghys, Audrey E. Pettifor.

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
