## [Editor Report · Decision Letter 0]

31 Jan 2020

Dear Dr Kilburn, 

Thank you very much for submitting your manuscript entitled "A Systematic Review of Cash Transfers for HIV Prevention: What do we know?" for consideration by PLOS Medicine.

Your manuscript has now been assessed by the PLOS Medicine editorial staff and I am writing to let you know that we would like to send your submission out for external peer review.

Kind regards,

Richard Turner, PhD

Senior editor, PLOS Medicine

rturner@plos.org

---

## [Editor Report · Decision Letter 1]

6 Feb 2020

Dear Dr Kilburn, 

Thank you for submitting your manuscript entitled "A Systematic Review of Cash Transfers for HIV Prevention: What do we know?" for consideration by PLOS Medicine.

Your manuscript has now been evaluated by the PLOS Medicine editorial staff and I am writing to let you know that we would like to send your submission out for external peer review.

However, before we can send your manuscript to reviewers, we need you to complete your submission by providing the metadata that are required for full assessment. To this end, please login to Editorial Manager where you will find the paper in the 'Submissions Needing Revisions' folder on your homepage. Please click 'Revise Submission' from the Action Links and complete all additional questions in the submission questionnaire.

Kind regards,

Richard Turner, PhD

Senior editor, PLOS medicine

rturner@plos.org

---

## [Decision Letter · Decision Letter 2]

16 Dec 2020

Dear Dr. Stoner,

Thank you very much for submitting your manuscript "A Systematic Review of Cash Transfers for HIV Prevention: What do we know?" (PMEDICINE-D-20-00282R2) for consideration at PLOS Medicine. We do apologize for the long delay in sending you a decision. 

Your paper was evaluated by a senior editor and sent to independent reviewers. One review is appended at the bottom of this email (a further report will be sent to you via email if/when it becomes available) and any accompanying reviewer attachments can be seen via the link below:

[LINK]

In light of these comments, we will not be able to accept the manuscript for publication in the journal in its current form, but we would like to invite you to submit a revised version that fully addresses the reviewer's and editors' comments. You will appreciate that we cannot make a decision about publication until we have seen the revised manuscript and your response, and we expect to pursue re-review, possibly involving additional reviewers. 

We hope to receive your revised manuscript by Jan 11 2021 11:59PM. Please email us (plosmedicine@plos.org) if you have any questions or concerns.

Please let me know if you have any questions. Otherwise, we look forward to receiving your revised manuscript in due course. 

Sincerely,

Richard Turner, PhD

rturner@plos.org

Please revisit the title, removing the rhetorical question and placing the study descriptor ("A systematic review") after a colon. 

Please remove information on funding and competing interests from the title page. In the event of publication, this information will appear in the article metadata, via entries in the submission form. 

Please adapt the abstract to a three-part format. The final sentence of the "Methods and findings" subsection should begin "Study limitations include ..." or similar and quote 2-3 of the study's main limitations. 

In the abstract, please note the countries or regions where the constituent studies were done; and state the proportions of different study designs included. 

After the abstract, we will need to ask you to add an "author summary" section in non-identical prose. You may find it helpful to consult one or two recent research papers in PLOS Medicine to get a sense of the preferred style. 

Please add a new final sentence to the "Introduction" section of the main text, stating the study's aim. 

Please quote the date of the search(es) in the Methods section. 

Please update the literature search to a date in the past 3 months.

In the results section and any other instances, please make that "sub-Saharan Africa". 

Throughout the paper, please take care in using words such as "effect" which imply causation. We generally restrict use of this language to describe evidence from randomized studies, and suggest alternatives such as "was associated with" or "impact" to describe findings from observational studies. 

Throughout the text, reference call-outs should be in the form of numbers formatted as follows: "... 130 countries [1].".

Please reformat the reference list so that entries appear in order of citation in the text. Citations should be in Vancouver format, and all italics should be converted into plain text. 

Please adapt the attached PRISMA checklist so that individual items are referred to by section (e.g., "Methods") and paragraph number rather than by line or page numbers, as the latter generally change in the event of publication. Please refer to the checklist in the methods section ("See S1_PRISMA_Checklist" or similar). 

Please also refer to the attached protocol document in the methods section, and note when this was prepared. Was this registered in any publicly-accessible database?

Comments from the reviewers:

*** Reviewer #1: 

Relevance and Interest

The article is relevant to a large audience of readers in both high-income countries and low-income settings. The study synthesizes well and provides a detailed narrative the compares and contrasts studies evaluating the effect of cash transfers on HIV and STI prevention. 

Impact

This article will have a medium-degree of impact. While the article does make mention of some other outcomes evaluated (such as school enrollment and health service utilization), the outcomes reported on in this systematic literature review seem limited. It misses some of the key outcomes evaluated by studies importance to HIV/STI prevention, such as ART adherence, given that adherence impacts community viral load and HIV testing patterns. While some of these studies are captured in the Bastagli et al review, some are missed.

Content

The narrative review of the findings is well reported. The introduction could flow better if the definition section on conditional vs. unconditional cash transfers came sooner and the section ended with the aim of the systematic review. In the results and discussion, it would be helpful to the reader if follow-up times for each study are reported and compared and contrasted. This is particularly important for studies that evaluated the effect of cash transfers on measures of disease incidence and prevalence. The outcomes table is difficult to read. It is advised that the authors consider synthesizing this information into a figure that displays important key findings. The last paragraph of the discussion presents quite a lot of information that has already been reported. It is advised that the authors consider making the concluding paragraph a discussion on why this matters for policy-makers and suggest a way forward for program planners that may be considering the implementation of cash transfers.

Originality

The article is original, but it also builds on a previous review by Bastagli et al. With understanding that there are many outcomes measured and with differences in measurement, was there any consideration given to how outcomes may be quantitatively meta-analyzed? This would add strength and greater originality to the manuscript.

***

[LINK]

---

## [Decision Letter · Decision Letter 3]

25 Apr 2021

Dear Dr. Stoner,

Thank you very much for submitting your revised manuscript "Cash Transfers for HIV Prevention: A Systematic Review" (PMEDICINE-D-20-00282R3) for consideration at PLOS Medicine. We do apologize for the delay in sending you a response. 

Your paper was seen by two further reviewers, including a statistical reviewer. The reviews are appended at the bottom of this email and any accompanying reviewer attachments can be seen via the link below:

[LINK]

In light of these reviews, we will not be able to accept the manuscript for publication in the journal in its current form, however we would like to invite you to submit a further revised version that addresses the reviewers' and editors' comments fully. You will recognize that we cannot make a decision about publication until we have seen the revised manuscript and your response, and we may seek re-review by one or more of the reviewers. 

We hope to receive your revised manuscript by May 14 2021 11:59PM. Please email us (plosmedicine@plos.org) if you have any questions or concerns.

Please let me know if you have any questions, and we look forward to receiving your revised manuscript. 

Sincerely,

Richard Turner, PhD

rturner@plos.org

Please resubmit your paper as a research article.

In your abstract, please quote the date of the most recent search rather than "the present".

In the Methods and Findings" subsection of your abstract, please adapt the presentation to the form: "... 20 (72%) were randomized trials ..."; and make that "... was a case-control study". Please make similar changes in the Main text.

At line 110, for example, please adapt the text to "... young women aged 13-22 years, and their families, to stay ...".

At line 166, for example, please quote 95% CI if available; similarly, if available please quote exact p values or "p<0.001".

At line 221, please make that "... for girls than boys".

Please hyphenate "quasi-experimental"; however "syphilis" should not take an initial capital.

Please italicize species names. 

Throughout the text, please style reference call-outs as follows: " ... STI negative [2,3].".

Noting reference 9 and others, please ensure that all references have full access information.

Please use the abbreviation "PLoS ONE".

Please include the "studies included" in the reference list (we think that some are already included but not all). 

Please rename the PRISMA attachment "S1_PRISMA_Checklist" or similar, and refer to it as such in the Methods section. 

You may wish to use the updated PRISMA, which has been published recently. 

Comments from the reviewers:

*** Reviewer #2: 

[See attachment]

Michael Dewey

*** Reviewer #3: 

This is a revision of a manuscript that provides a systematic review of studies evaluating cash transfers on a range of health outcomes, including HIV and STIs and sexual behaviours that potentially impact on HIV risk. The manuscript is relevant and of interest to HIV prevention researchers. The authors have addressed the previous reviewer's comments. I have some additional minor comments.

1. The authors may want to consider changing the title. The scope of the review covers more than just HIV prevention and several of the studies do not assess HIV outcomes

2. Page 6, line 55 - there is a superscript "h" in the middle of the sentence

3. Page 9, line 127 - define HSV-2 at first use. It is written in full in line 139

4. Page 10, line 155, 156, 159 - there are a couple of extra brackets

5. Page 11, line 175 - the Minnis et al ref needs to be provided in Vancouver style

6. Page 11, line 176 - the one bracket "(" after the references (25-29) needs to be removed

7. The point about the limited data in key populations is made several times in the discussion, e.g. page 14, line 248, page 15, line 273 and 276 - some of the redundancy can be removed

8. References - There is some inconsistency with the formatting of the references. Ref 2 for example, ref 9 is missing volume and page numbers, provide url for ref 27 if available, ref 39 has "Suppl" three times

9. Table 2 - this table contains a lot of information and is not easy to follow. Not all outcomes are assessed in each study. Perhaps a not assessed (NA) can be added to the cells where the data are not assessed.

***

[LINK]

---

## [Decision Letter · Decision Letter 4]

25 Oct 2021

Dear Dr. Stoner,

Thank you very much for re-submitting your manuscript "Cash Transfers for HIV Prevention: A Systematic Review" (PMEDICINE-D-20-00282R4) for consideration at PLOS Medicine. We do apologize for the delay in our response.

I have discussed the paper with our academic editor and it was also seen again by one reviewer. I am pleased to tell you that, provided the remaining editorial and production issues are fully dealt with, we expect to be able to accept the paper for publication in the journal.

[LINK]

We hope to receive your revised manuscript within 2 weeks. Please email us (plosmedicine@plos.org) if you have any questions or concerns.

Please let me know if you have any questions, and we look forward to receiving the revised manuscript.   

Sincerely,

Richard Turner, PhD

rturner@plos.org

Requests from Editors:

Noting referee 2's comments, we ask you to do an additional search using this methodology and incorporate the findings in a supplementary file, adding a paragraph, say, to the results section to quote the findings. 

Please restructure the abstract: the current "Background" sentence should begin the "Methods and findings" subsection. We suggest crafting a new "Background" subsection of at least two sentences, aiming to explain why HIV prevention is important, for example, and the relevance of cash transfers in this area.

Where you quote "45 publications" in the abstract, please add a few additional words to quote the number of studies screened and excluded, for example.

In the abstract, please make that "observational studies", "case-control study" etc, and make similar amendments in the results section (main text).

Please state the reason for not doing a meta-analysis in the abstract.

Where you refer to a "large effect of the program" in the abstract, for example, please confirm that this refers to evidence from randomized trials. We generally ask authors not to use language implying causality from weaker study designs. 

Regarding apparent program effects on sexual debut, different numbers seem to be quoted in the abstract ("10/18" studies), author summary ("11/18") and main text (line 185; "9/18"). Please check the numbers and report consistent findings throughout.

Early in the author summary we suggest "... a ... strategy".

We think that should be "December 2020" in the author summary.

At line 76, please state the year for "February 4th".

Should "that" be removed at line 105?

At line 116, should that be "HIV incidence or prevalence ..."?

Noting "p<0.05" at line 121, please quote exact p values where available or, for smaller values, "p<0.001" throughout the ms.

At line 132, please amend the wording to "... found no significant effect on HIV prevalence ... although there was a non-significant reduction in one study" or similar.

At line 143, please amend the current wording ("... smaller effect on syphilis") to indicate that the apparent reduction was not statistically significant (e.g., "... reduced prevalence of HSV-2 but not of syphilis.").

At line 159 and any other instances in the ms, please substitute "sex" for "gender" where appropriate.

Again at line 170 (RR 0.86, 95% CI 0.67-1.1) please amend the wording to indicate more clearly that the apparent effect was not statistically significant.

At line 182, can you add a few words to give the reader an idea what "larger reduction" was investigated in the study/ies in question?

At lines 204-6, where you discuss the findings of an observational study, please adapt the language to avoid implying causality, e.g., "... HIV risk behavior were lower than with receipt of cash alone.".

At line 215, please make that "In this systematic review ...".

At line 243, please avoid "incredibly".

At line 287, should that be "we were unable ..."?

Please use the general style "... aged 18 years" throughout.

We suggest hyphenating "cash-transfer" throughout when used as an adjective.

Please use the journal name abbreviation "PLoS ONE" in the reference list.

Please provide additional information for reference 69. Is a URL available, for example?

Comments from Reviewers:

*** Reviewer #2: 

The authors continue to be reluctant to include studies not in English. They might be interested in the article by Jackson, J L and colleagues "The accuracy of Google Translate for abstracting data from non--English--language trials for systematic reviews." Annals of Internal Medicine 2019 (171) 677-679. Not perfect but better than dropping them altogether.

Michael Dewey

***

[LINK]

---

## [Editor Report · Decision Letter 5]

8 Nov 2021

Dear Dr. Stoner,

Thank you very much for re-submitting your manuscript "Cash Transfers for HIV Prevention: A Systematic Review" (PMEDICINE-D-20-00282R5) for consideration at PLOS Medicine.

I have discussed the paper with our academic editor, and we will need to ask you to address some additional points before we are in a position to proceed further. 

The remaining issues that need to be addressed are listed at the end of this email: please take these into account before resubmitting your manuscript.

In revising the manuscript for further consideration here, please ensure you address the specific points made by the editors. In your rebuttal letter you should indicate your response to the editors' comments and the changes you have made in the manuscript. Please submit a clean version of the paper as the main article file. A version with changes marked must also be uploaded as a marked up manuscript file.

Please let me know if you have any questions in the meantime, and we look forward to receiving the revised manuscript.   

Sincerely,

Richard Turner, PhD

rturner@plos.org

Requests from Editors:

Please avoid "almost a decade" in the abstract. We suggest "several years".

We suggest quoting some additional findings from the study in the abstract: the observations in the two summary points beginning "Eight of the 27 studies ..." and "Of the studies ..." could be included (bearing in mind that the abstract will appear in Pubmed, whereas the Summary points will not). 

Once this is done, please adapt the summary points to avoid repetition.

Requests from Academic editor:

An issue that needs to be resolved is the inconsistency in the number of studies included / excluded.

1. There is some inconsistency in the number of studies included - Line 254 and 264 refers to 25 studies included in this review but the abstract gives 27 studies. I think it should be 27 – table 1 lists 27 studies

2. The numbers in the consort don’t quite add up – In figure 1 there are 78 studies included and then 23 are excluded – that would mean there were 55 studies remaining but the consort shows 45. In the text it states “78 studies were included and 45 were excluded that did not meet inclusion criteria. We identified a total of 45 peer reviewed publications” – if 45 were excluded then it would leave 33 publications not 45.

Minor comments – 

3. authors should not refer to Table 1 in the abstract

4. extra full stop in line 223

5. Figure 1 – the reason for excluding the 1484 papers should be provided. Does the 1642 include the non-English studies? 

***

---

## [Editor Report · Decision Letter 6]

11 Nov 2021

Dear Dr Stoner, 

On behalf of my colleagues and the Academic Editor, Dr Baxter, I am pleased to inform you that we have agreed to publish your manuscript "Cash Transfers for HIV Prevention: A Systematic Review" (PMEDICINE-D-20-00282R6) in PLOS Medicine.

PRESS

Sincerely, 

Richard Turner, PhD 

rturner@plos.org